# Arbuscular Mycorrhizal Fungi Reduce Cadmium Leaching from Sand Columns by Reducing Availability and Enhancing Uptake by Maize Roots

**DOI:** 10.3390/jof8080866

**Published:** 2022-08-17

**Authors:** Zihao Yu, Xiaoling Zhao, Xinran Liang, Zuran Li, Lei Wang, Yongmei He, Fangdong Zhan

**Affiliations:** 1College of Resources and Environment, Yunnan Agricultural University, Kunming 650000, China; 2College of Horticulture and Landscape, Yunnan Agricultural University, Kunming 650201, China

**Keywords:** arbuscular mycorrhizal fungi, root exudates, Cd distribution, Cd migration, sand column

## Abstract

To explore the effect of arbuscular mycorrhizal fungi (AMF) on the environmental migration of cadmium (Cd), a sand column-maize system containing 20 mg·L^−1^ Cd solution was used to investigate the AMF effect on maize growth, Cd uptake by maize, Cd adsorption by sand and Cd leaching loss. The results showed that AMF significantly increased the content of EE-GRSP and T-GRSP by 34.9% and 37.2%, respectively; the secretion of malonic acid, oxalic acid and succinic acid increased by 154.2%, 54.0% and 11.0%, respectively; the secretion of acetic acid and citric acid increased by 95.5% and 59.9%, respectively; and the length, surface area, volume, tip number and cross number of maize roots decreased by 10%, 15%, 17%, 20% and 36.4%, respectively. AMF significantly increased Cd adsorption by sand by 6.2%, Cd uptake by maize by 68.1%, and Cd leaching loss by 84.6%. In the sand column-maize system, 92.3% of the total Cd was adsorbed by sand, 5.9% was taken up by maize and 1.8% was lost due to leaching. Moreover, Cd adsorption by sand was significantly positively correlated with the GRSP content and oxalic acid secretion, and Cd uptake by roots was significantly negatively correlated with Cd leaching loss. Overall, AMF reduced the loss of Cd in the leaching solution by promoting the release of oxalic acid and GRSP, increasing the adsorption of Cd in the sand and fixing the Cd in the plant to the roots.

## 1. Introduction

Due to industrial and agricultural production activities such as metal mining and smelting, sewage irrigation and sludge application, the pollution of heavy metal cadmium (Cd) in farmland soil has become increasingly serious and one of the major environmental problems in the world [1]. The majority of the Cd that enters the soil is absorbed by soil particles and eventually generates a Cd pool in the environment [2]. However, the activity and migration ability of Cd is strong [3]. Soil Cd easily migrates with water flow generated by precipitation and irrigation [4] and is taken up by plant roots [5]. The environmental migration of Cd derived from soil Cd pools, such as leaching and loss with water flow and uptake by plants, destroys the permeability of the cell membrane, makes the binding enzymes and other substances on the membrane dysfunctional, accelerates the extravasation of the substances needed for plant growth, causes the physiological and biochemical processes in the body to be disordered, changes the hormone levels and nucleic acid metabolism levels in plants, and accelerates plant death [6,7]. Therefore, the problem of Cd leaching and loss in contaminated soil has attracted widespread attention.

Arbuscular mycorrhizal fungi (AMF) are widely distributed soil fungi that can form symbiotic relationships with approximately 71% of angiosperms (including most crops) [8,9]. According to Janouková and Pavlková [10], AMF spread out and expand in the soil to produce dense hyphal networks, the hyphae wrap around the soil particles, and there are binding sites on the hyphae that can adsorb and fix Cd, reducing the loss of Cd. Hyphae secrete a protein called glomalin-related soil protein (GRSP), which can alter the surface characteristics of soil particles and increase Cd adsorption [11]. AMF can also boost the nutrition and growth of the host plant, promote root growth and the secretion of low molecular organic acids, further promote the chelation and adsorption of Cd, reduce the toxicity of Cd and play an important role in soil Cd fixation [12,13,14,15]. AMF significantly affects the Cd distribution in plants; some studies found that Cd was mainly fixed in AMF-inoculated roots, and Cd transport to shoots was reduced, resulting in less Cd uptake in shoots [16,17,18]. In addition, AMF reduces soil Cd leaching loss under rainfall conditions [19]. Therefore, AMF significantly affects the environmental migration of Cd in soil-plant systems.

Because most of the previous studies used polluted soil as the matrix, the soil composition was complex, which made the quantitative analysis of Cd migration in the soil-plant system difficult to control. Amir et al. [20] found that although AMF could promote Cd adsorption by soil, this effect was subtle. The influence of the minor changes in Cd adsorption in soil induced by AMF on environmental Cd migration, such as plant Cd uptake and Cd leaching loss, is still unclear. Replacing the soil matrix with quartz sand with simple composition can effectively reduce the interference of complex soil composition and is often used in quantitative experimental studies to control a single variable [21].

Due to the large biomass, developed root and certain Cd tolerance, maize is often used in the study of AMF on crop growth and Cd absorption under Cd stress [22]. Therefore, in the present study, the experiment using sand culture with Cd treatment was carried out to study the effect of AMF inoculation on maize growth, Cd content and Cd migration in the sand column-maize system. We expected that AMF would have a minor impact on sand Cd adsorption, but would have a significant impact on plant Cd absorption and Cd leaching loss, as well as modify the Cd distribution in the sand column-maize system.

## 2. Material and Methods

### 2.1. Test Materials

The quartz sand (the main component of SiO_2_) used in the experiment had particle sizes of 16–32 mesh (0.5–1 mm) and 200–300 mesh (0.05–0.07 mm). To eliminate surface adsorption, the quartz sand was immersed in a 10% nitric acid solution for 24 h. Substances were cleaned in distilled water before being autoclaved for 2 h at 121 °C.

The tested maize (*Zea mays*) variety was Huidan No. 4, which was a low-accumulation maize variety screened by the research group [23]. Maize seeds were soaked in 10% sodium hypochlorite solution for 2 min, removed, washed, placed in 75% ethanol solution for sterilization for 1 min, rinsed with distilled water, and placed in a sterilized Petri dish padded with soaked filter paper. The cells were cultivated in a constant temperature incubator at 28 °C for 3 days. When the seeds germinated and sprouted white buds, seedlings with no pollution and consistent growth were selected for use.

The strain used in this study was *Funneliformis mosseae*, which is highly tolerant to Cd and was provided by the Beijing Academy of Agriculture and Forestry’s Institute of Plant Nutrition and Resources (BGCYN05, 1511C0001BGCAM0013) [24]. Hoagland solution was prepared from modified Hoagland formula, iron salt solution and trace element solution [25]. CdCl_2_·2.5H_2_O (2.04 g) was dissolved in 10 L of Hoagland nutrient solution to prepare a Cd-containing Hoagland nutrient solution with a Cd concentration of 20 mg·L^−1^.

### 2.2. Sand Column Preparation and Leaching Experiment

The leaching device’s bottom bracket was stainless steel, while the upper half was a cuboid frame with welded brackets at the bottom four corners, and the ground was raised by 15 cm (Figure 1A). The leaching column was a hollow polyvinyl chloride cylinder with an open-top, a height of 35 cm, and a radius of 5.5 cm. The hollow cylinder was opened at 13 cm and 23 cm, respectively, from top to bottom, with the intermediate position at the bottom. At the start of the project, a soil solution sampler (Rhizon MOM, Shanghai Qiwen Biology Science and Technology Co., Ltd., Shanghai, China) was installed.

A 400-mesh (38 μm) nylon cloth was laid at the bottom of the leaching column, and the bottom was filled with 2 cm thick coarse quartz sand. Then, the wet method [26] was used to fill the fine quartz sand. Two treatments, the control group (CK) and AMF inoculation group (AMF), were established, and each treatment was repeated four times. A total of 200 g of strain was laid on the surface of the AMF-treated sand column, covered with 3 cm of fine quartz sand after planting maize and filled with sterile quartz sand for the control group (Figure 1B). Four seeds with the same growth degree were planted in each leaching column, and after 7 days of planting, only one maize seedling with good growth was left in each leaching column. After 3 days of seedling thinning, 2 L of Hoagland solution containing Cd at concentrations of 0 and 20 mg L^−1^ was applied to each leaching column. Cd-free Hoagland solution was added every other day to keep the volume of the sand column solution constant.

The leaching experiment began after 60 days of maize growth. Each leaching column was gradually filled with 1.3 L of distilled water. The sand column solution was collected after 10 days using a soil midstream sampler at 10 and 20 cm sand column depths, with the valve placed at the bottom of the sand column. After that, the eluent was collected from the sand column.

### 2.3. Determination of AMF Colonization

After the leaching experiment (maize was grown for 72 days), maize root samples and rhizosphere quartz sand were collected to determine the infection characteristics of AMF. The fresh root samples were cut into approximately 1 cm root segments, and after alkaline dissociation and blue ink dyeing, the infection rate of AMF was determined by the improved method of triphenyl blue staining [27].

The AMF spore number was determined by sucrose centrifugation [28]. Rhizosphere quartz sand (10.0 g) was weighed and washed repeatedly with a 0.25 mm sieve, and the residue of the sieve was transferred to a 100 mL centrifuge tube and centrifuged at 3000 r·min^−1^ for 10 min. After removing the supernatant, 50% sucrose solution was added, fully shaken, and centrifuged at 3000 r·min^−1^ for 10 min to obtain the supernatant, and the supernatant was filtered with filter paper. The number of AMF spores on the filter paper was observed under a stereo microscope (SZ61, Thermo Fisher Scientific, Shanghai, China).

Two indoor air-dried rhizosphere quartz sand samples, 1 g each, were extracted with 20 mmol·L^−1^ sodium citrate buffer (pH 7.0) one time (extraction time 30 min) and 50 mmol L^−1^ sodium citrate buffer (pH 8.0) five times (extraction time 60 min each time) at 121 °C and 0.11 easy extractable glomalin-related soil protein (EE-GRSP) and total glomalin-related soil protein (T-GRSP) solutions were obtained after centrifugation at 6000 r·min^−1^ for 15 min, and the GRSP content in the supernatant was determined by the Bradford method [29].

The density of hyphae was measured by the vacuum pump microporous membrane suction filtration method [30]: 5 g of the rhizosphere sand sample was tested in a 500 mL beaker, 250 mL of water and an appropriate amount of soil dispersant (sodium hexametaphosphate) were added, and after standing for 5 min and fully stirring with a glass rod, the suspension was passed through multilayer sieves with different apertures and washed with distilled water. The substances were rinsed in the lower sieve into a mechanical stirrer, stirred for 30 s, then the suspension was transferred to a triangular flask, 10 mL (twice, 5 mL each time) was sucked into a 0.45μm microporous filter membrane for vacuum pumping, a few drops of triphenyl blue solution were added for dyeing, and the intersection of 25 fields of vision were observed under a microscope (Leica DM 2000 LED, Leica Microsystems, Wetzlar, Germany) after the filter membrane was slightly dried, and then analysed the by grid crossing method.

### 2.4. Determination of Root Exudates, Root Morphology and Plant Biomass

After collecting rhizosphere samples, maize roots were washed, and maize root exudates were collected by the hydroponic collection method [31]: the leaned maize roots were placed in distilled water and cultured in a dark room for 24 h. After the culture solution was suction filtered, it was concentrated by a vacuum rotary evaporator at 45 °C for testing. The organic acids secreted by roots were analysed by high-performance liquid chromatography [32]. The standard acids were succinic acid, malic acid, malonic acid, acetic acid, citric acid and tartaric acid. The measurement conditions were as follows: the model of HPLC was Agilent 1100, the models of separation column and protection column were IonpacAS11-HC and Ion pac AS11-HC, the model of anion microfiltration inhibitor was ASRS-11, the eluent was 2% methanol, 98% water (containing 0.4% phosphoric acid, etc.), the column temperature was 30 °C, and the eluent flow rate was 0.6 mL·min^−1^.

The roots were separated from the shoots, placed in a colorless transparent water tank filled with deionized water, the position of the roots was adjusted with tweezers to avoid overlapping, the complete root image was scanned with a root scanner (Epson perfection V800, Nagano Prefecture, Japan), and then WinRhizo PRO 2016 root analysis system software matched with the scanner was used to obtain the average root diameter, root length, root volume, root tip number, branch number and root surface area.

EDTA solution was used to clean maize roots and shoots to remove Cd adsorbed on the surface, and then rinsed with distilled water. The roots and shoots were placed in an electric blast drying oven (101-0BS, Lichen Instrument, Zhejiang, China) at 105 °C for 30 min and then dried to constant weight in a 75 °C oven, and the biomass of the roots and shoots was obtained by weighing.

### 2.5. Determination of Cadmium Concentration, Uptake by Plants and Leaching Loss

A 0.250 g maize plant sample was screened by a 100 mesh sieve, 10 mL of nitric acid was added, and then put into an electrothermal digestion apparatus (ED54, Lab Tech, Beijing, China) for digestion. After digestion at 80 °C for 1.5 h, 120 °C for 1.5 h, and 150 °C for 2 h, the temperature was raised to 175 °C to remove the acid to approximately 1 mL, and the mixture was rinsed with 1% nitric acid several times. ICP-MS (CAP Q, Thermo, Waltham, MA, USA) was used to determine the Cd content in the solution to be tested.

The sand column solution and the leaching solution were collected and stored in plastic bottles, and 25 mL of each was taken to determine the Cd concentration by HNO_3_-H_2_O_2_ digestion-graphite furnace atomic absorption spectrometry.

### 2.6. Determination of Cadmium Adsorption by Inner Wall of Column and Sand

A total of 1 L of 6 mg·L^−1^ nitric acid was used to clean the inner wall of the sand column to elute the Cd adsorbed by the inner wall of the sand column, and the Cd concentration of the solution was measured in a graphite furnace. The nitric acid, hydrochloric acid, formaldehyde, glacial acetic acid, ethanol, and hydrogen peroxide used in this experiment were guaranteed reagents. The glassware and the pressure digestion tank used for digestion were soaked in 25% nitric acid solution for 24 h before use, and CdCl_2_ was used as the standard solution of the appropriate quality.

Two grams of air-dried quartz sand was weighed and put into a 100 mL centrifuge tube, 50 mL of 0.1 mg·L^−1^ NaNO_3_ solution was added, and then the pH of the suspension was adjusted to 5.0 using dilute HNO_3_. The cells were held at a constant temperature overnight after high-speed shaking at 180 rpm at 25 °C for 2 h in a constant temperature shaker and centrifuged at 2574 g for 10 min, and the supernatant was preserved. A flame atomic spectrophotometer was used to detect the concentration of the metal ion Cd (AA-3600, Shanghai Yuanyan Instrument Co., Ltd., Shanghai, China).

### 2.7. Data Processing and Statistical Analysis

The proportion of Cd content in each part of the sand column system (%) = the amount of Cd in each part (mg)/the amount of Cd in the system (mg) ×100.

The test data were handled in Excel 2016 and the average of four replicates was taken, with the mean and standard deviation reported as the mean and standard deviation. OriginPro was used to create charts, SPSS was used to perform single-factor ANOVA and independent samples *t* tests, and LSD and Duncan models were utilized to conduct multiple group comparisons. Pearson correlation analysis was performed on GRSP, root morphology and exudate, maize Cd uptake, Cd adsorption in quartz sand, Cd concentration in sand column solution and Cd leaching loss.

## 3. Results

### 3.1. Colonization of Arbuscular Mycorrhizal Fungi in the Maize Rhizosphere

Under 20 mg·L^−1^ Cd stress, the root AMF infection rate was 27.5%, the number of quartz sand spores per gram dry weight was 42, and the mycelial density was 1.1 m·g^−1^; Inoculation with AMF significantly increased the content of EE-GRSP and T-GRSP by 34.9% and 37.2%, respectively (Figure 2). This result indicated that AMF and maize successfully established a symbiotic relationship.

### 3.2. Effect of Arbuscular Mycorrhizal Fungi on Maize Biomass, Root Morphology and Root Exudate Content

Inoculation with AMF significantly decreased the biomass of the shoots and roots by 50.3 and 30.0%, respectively, under the 20 mg·L^−1^ Cd treatment (Figure 3). It is clear that AMF inoculation prevents maize from growing.

Inoculation with AMF reduced root length, root volume, root surface area, root tip number, and root crossing numbers by 16.7%, 20.0%, 17.0%, 10.5% and 36.4%, respectively (Table 1), while increasing malonic acid, succinic acid, and oxalic acid content by 154.2%, 11.0% and 54.0%, respectively, and decreasing acetic acid and citric acid content by 95.5% and 59.9%, respectively (Table 2). It was discovered that inoculating maize roots with AMF affected the concentration of low molecular organic acids in the roots and reduced root morphology.

### 3.3. Effect of Arbuscular Mycorrhizal Fungi on Cadmium Content and Uptake by Maize

Inoculation with AMF significantly reduced the Cd content and uptake in shoots under the 20 mg·L^−1^ Cd treatment by 53.8% and 24.6%, respectively, and significantly increased root Cd content and uptake under the 20 mg·L^−1^ Cd treatment, with increases of 78.0% and 123.1%, respectively; the total Cd uptake in maize plants increased by 68.1% (Figure 4). AMF can promote the uptake of Cd in plants but inhibit the transport of Cd from roots to shoots.

### 3.4. Effect of Arbuscular Mycorrhizal Fungi on Cd Concentration in Sand Solution and Cd Leaching Loss

The inoculation of AMF significantly reduced the solution Cd concentration and Cd leaching loss (Figure 5). This shows that AMF has the ability to reduce Cd leaching loss.

### 3.5. Effect of Arbuscular Mycorrhizal Fungi on Cd Adsorption by Sand

Under the treatment of 20 mg·L^−1^ Cd, Cd adsorption by sand without inoculation with AMF was 35.4 ± 0.5 mg·kg^−1^, and Cd adsorption by sand under inoculation with AMF was 37.6 ± 0.6 mg·kg^−1^, an increase of 6.2%.

### 3.6. Effect of Arbuscular Mycorrhizal Fungi on Cadmium Distribution in Sand Column-Maize System

The proportion of Cd in each part of the sand column system from high to low is Cd adsorption by sand, Cd uptake in maize, and Cd leaching loss. AMF inoculation significantly increased Cd adsorption by sand by 6.2% and significantly decreased Cd leaching loss and Cd uptake in maize by 84.6% and 54.9%, respectively (Table 3). The sand column acts as a Cd pool, and slight changes in its content lead to significant changes in Cd loss (plant uptake, leaching), indicating that AMF have a significant effect on Cd adsorption.

### 3.7. Correlation Analysis

Correlation analysis indicated that root length, root surface area, root volume, root tip number, and crossover number were positively correlated with shoot Cd content and uptake and Cd leaching loss and negatively correlated with GRSP, root Cd content and uptake (Table 4). In addition, Cd leaching loss was significantly negatively correlated with the uptake of Cd in shoots (r = 0.967 *p* < 0.01, n = 8) and was significantly negatively correlated with the uptake of Cd in roots (r = −0.878, *p* < 0.01, n = 8). AMF could affect root morphology by promoting the release of GRSP and then affecting the uptake of Cd and Cd leaching loss.

Correlation analysis indicated that T-GRSP and EE-GRSP were significantly negatively correlated with acetic acid and citric acid, and EE-GRSP was significantly positively correlated with oxalic acid, succinic acid, and malonic acid. Oxalic acid was significantly positively correlated with Cd adsorption by sand, and the latter was significantly negatively correlated with acetic acid and citric acid (Table 5). In addition, Cd adsorption by sand was significantly positively correlated with EE-GRSP (r = 0.692, *p* < 0.05, n = 8) and was significantly positively correlated with T-GRSP (r = 0.761, *p* < 0.05, n = 8), Cd adsorption by sand was significantly negatively correlated with Cd leaching loss (r = −0.746, *p* < 0.05, n = 8). It shows that AMF can affect the content of low-molecular-weight organic acids secreted by roots by promoting the release of GRSP and then affecting Cd adsorption by sand.

Correlation analysis showed that Cd uptake in maize shoots was significantly positively correlated with the root surface area (r = 0.575, *p* < 0.01) and root volume (r = 0.663, *p* < 0.01), with the concentrations of root tartaric acid (r = 0.418, *p* < 0.05), malic acid (r = 0.448, *p* < 0.05) and succinic acid (r = 0.428, *p* < 0.05) secreted by the roots being positively correlated. Thus, the increase in Cd uptake in maize caused by AMF is closely related to the improvement of root morphology and LMWOA secretion.

## 4. Discussion

### 4.1. Effect of Arbuscular Mycorrhizal Fungi on Cadmium Adsorption by Sand

The soil Cd pool is a Cd repository, and its changes will affect the reactivity, mobility, and bioavailability of trace elements, thereby altering biodiversity, plant metabolism, and physiological processes [33]. The GRSP produced by the decomposition of AMF hyphae can enhance soil stability and has a strong binding ability to Cd, thus promoting the adsorption capacity of soil for Cd [34,35,36]. In addition, inoculation with AMF can promote the release of low molecular weight organic acids from roots, reduce soil pH, change the form of Cd, promote the chelation and adsorption of Cd, and affect the fixation of Cd in soil [37]. This is consistent with the conclusion found in this experiment that inoculation with AMF and Cd stress significantly increased the contents of malonic acid, oxalic acid, succinic acid and malic acid. This experiment also found that EE-GRSP has a very significant positive correlation with oxalic acid, succinic acid, and malonic acid, and oxalic acid has a very significant positive correlation with Cd adsorption by sand, indicating that AMF can promote Cd adsorption by sand by promoting root exudates. This is because organic acid molecules can chemically react with free Cd ions to generate stable compounds that exist in soil solutions and promote soil desorption of Cd [38]. However, this experiment also found that T-GRSP and EE-GRSP are significantly negatively correlated with acetic acid and citric acid, and Cd adsorption by sand is significantly negatively correlated with acetic acid and citric acid. The reason for this phenomenon is the biological characteristics of the strain itself or the inconsistency of the affinity between the strain and the host plant, and additionally, the differences in the soil environment and the physiological and biochemical characteristics of the plant itself can also cause this phenomenon [39].

### 4.2. Effect of Arbuscular Mycorrhizal Fungi on Plant Growth and Cadmium Uptake

The crop root is the part of the plant that comes into direct contact with the soil and is the principal organ for plant uptake, and its development is frequently influenced by Cd toxicity [40]. AMF inoculation has been demonstrated to enhance the root absorption range, increase the root-to-soil contact area, and promote root growth [41]. However, another study also found that if there is enough soil fertility in the growth medium to meet the needs of the roots, then the AMF hyphae will have difficulty obtaining nutrients to promote plant growth, thereby inhibiting the growth of the host plants [9]. Therefore, in our study, inoculation with AMF significantly reduced root volume, root tip number and root crossings number and reduced root and shoot biomass. AMF symbionts produce a huge mycelial network, which requires a large amount of nutrients to maintain their own growth and development and forms a part of the competition relationship with maize plants in nutrient absorption, which does not meet the growth and development needs of maize plants [42]. In addition, AMF inoculation leads to an increase in Cd content in maize roots, and the uptake of toxic Cd ions could mask the positive effect of mycorrhizae and change maize root characteristics, thereby inhibiting maize growth [43].

Cd uptake can reflect the absorption capacity of plants to soil Cd and can also reflect the degree of Cd stress in plant growth and development, that is, the tolerance of plants to Cd stress. In Cd-contaminated soil, AMF can play the role of filtration and chelation, adsorb many Cd through a large mycelial network, or “compartmentalize” Cd through structures such as arbuscules and vesicles and fix them in the symbiotic interface to inhibit the transport ability of Cd in plants and further weaken the toxic effects of Cd. At the same time, the inoculation of AMF can enrich the Cd ions at the binding sites of hyphae through the physical adsorption, coordination and microprecipitation of metal ions on the outer surface of cells and the binding effects of hyphae cell walls and intracellular thermostable proteins, polyphosphoric acid and organic acids, thus enhancing the adsorption performance and ability of plants to Cd [44]. In this experiment, it was found that inoculation with AMF also promoted the Cd uptake of plants but decreased the transfer of Cd to shoots, which also alleviated the damage of Cd to shoots. This is because inoculation with AMF will fix Cd in the vacuoles or cell walls of plant roots, reduce the accumulation of Cd in stems and leaves, change the absorption and distribution of Cd in maize, and hinder the transfer of Cd to stems and leaves [45,46]. However, some studies have found that inoculation with AMF reduced the Cd uptake of plants [47], which is because GRSP precipitates Cd in the soil through binding transformation, reducing the available state of Cd in soil and reducing the uptake of Cd by plants [8]. The effects of AMF inoculation on the Cd uptake of plants also had different results depending on the plant species, soil pollution, AMF species and their different combinations. In addition, the change in soil pH caused by root exudates also affects the content of Cd in solution and the content of available Cd in soil, which in turn affects Cd leaching loss and Cd uptake in plants [18].

### 4.3. Effect of Arbuscular Mycorrhizal Fungi on Cadmium Leaching Loss

Leaching Cd from the soil is a complicated process. In addition to crop roots, other factors, such as soil physicochemical qualities, soil texture, pollution levels, and others, might influence Cd leaching [48,49]. The matrix in this experiment was pure quartz sand, and exogenous Cd (CdCl_2_·2.5H_2_O) was introduced to the quartz sand column, with no influence on Cd loss. There are multiple migratory pathways for Cd after it enters the sand column-maize system: sand adsorption, maize absorption, and leaching loss. In this experiment, the content of Cd applied to the sand column-maize system was known. After maize growth and leaching, the Cd content and proportion of each part can be determined to clarify the effect of AMF on the distribution of Cd in the maize system. It was found that the proportion of Cd in the sand column-maize system decreased from high to low, followed by Cd adsorption by sand > Cd uptake in maize > Cd leaching loss, and inoculation with AMF reducing the Cd leaching loss by 85.17%, indicating that AMF played a significant role in reducing Cd leaching loss. In the experiment, it was found that Cd leaching loss was significantly negatively correlated with the uptake of Cd in shoot and was significantly negatively correlated with the uptake of Cd in the roots, indicating that AMF reduced the amount of Cd leaching loss by promoting the uptake of Cd by plants [50]. Second, AMF acts on soil particles through hyphae to form the skeleton of soil aggregates and further forms microaggregates and macroaggregates, which improves the soil structure and reduces Cd leaching loss.

In summary, AMF play a significant role in immobilizing Cd and reducing their migration from leaching from contaminated soils. The subtle changes in the soil Cd pool were closely related to environmental Cd migration (Cd uptake by plants and solution Cd loss). However, this experiment was an indoor sand culture experiment, and the process of crop growth and leaching is affected by many environmental factors. It is necessary to further study the influence of AMF on the leaching of contaminated soil under different environmental conditions and its mechanism. In addition, the forms of Cd in leaching solution also need to be further studied.

## 5. Conclusions

This study showed that AMF inoculation plays a significant role in reducing the leaching of Cd from contaminated sand. The subtle changes of Cd adsorption by sand were closely related to environmental Cd migration (Cd uptake by plants and solution Cd loss). However, further studies are necessary to determine the influence of AMF on the leaching of cadmium from contaminated soil under different environmental conditions.

## Figures and Tables

**Figure 1 jof-08-00866-f001:**
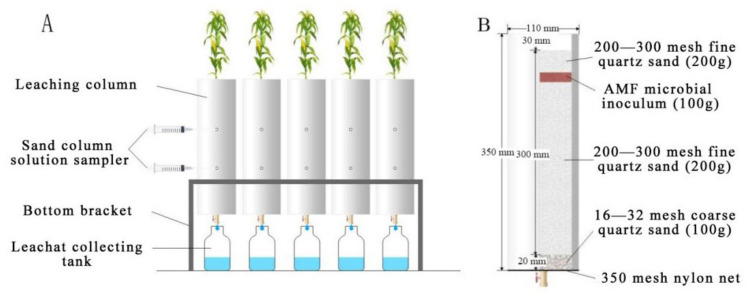
Filling and leaching of a sand column schematic diagram: (**A**) Schematic of the leaching device; (**B**) Schematic diagram of leaching column.

**Figure 2 jof-08-00866-f002:**
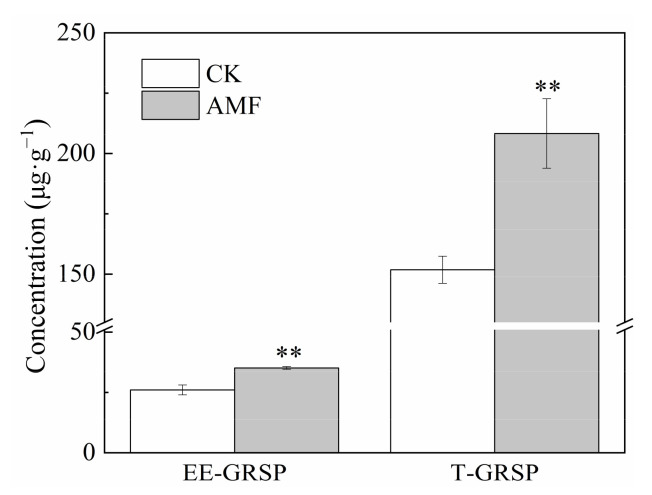
Effects of AMF on the content of EE-GRSP and T-GRSP under 20 mg·L^−1^ Cd concentration stress. Error bar indicates standard deviation, n = 4. EE−GRSP represents easily extractable glomalin-related soil protein, T−GRSP represents total glomalin-related soil protein, CK represents the control of non-inoculation, Cd represents cadmium. AMF represents *Funneliformis mosseae* inoculation. “**” mean *p* < 0.01 according to LSD test.

**Figure 3 jof-08-00866-f003:**
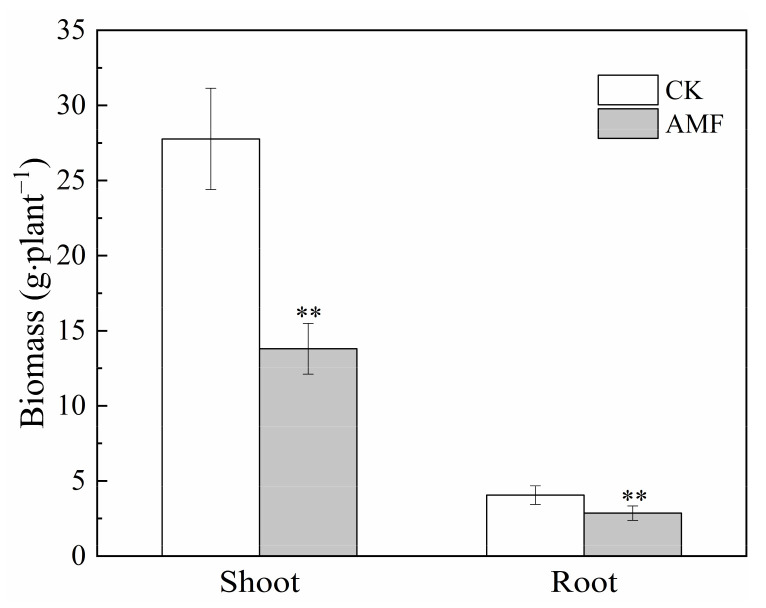
Effects of AMF on biomass of maize under 20 mg·L^−1^ Cd concentration stress. Error bar indicates standard deviation, n = 4. CK represents the control of non-inoculation, Cd represents cadmium. AMF represents *Funneliformis mosseae* inoculation. “**” mean *p* < 0.01 according to the LSD test.

**Figure 4 jof-08-00866-f004:**
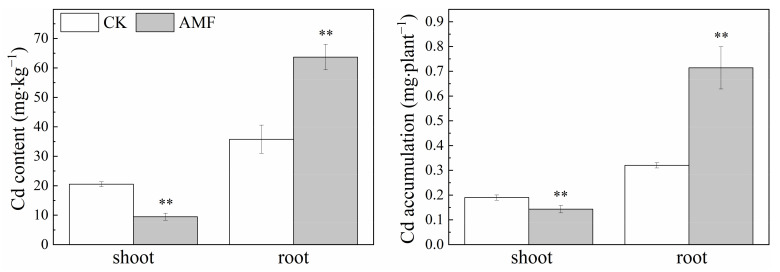
Effects of AMF on Cd content and uptake in maize plant under two Cd concentration stress. Error bar indicates standard deviation, n = 4. CK represents the control of non−inoculation, Cd represents cadmium. AMF represents *Funneliformis mosseae* inoculation. “**” mean *p* < 0.01 according to the LSD test.

**Figure 5 jof-08-00866-f005:**
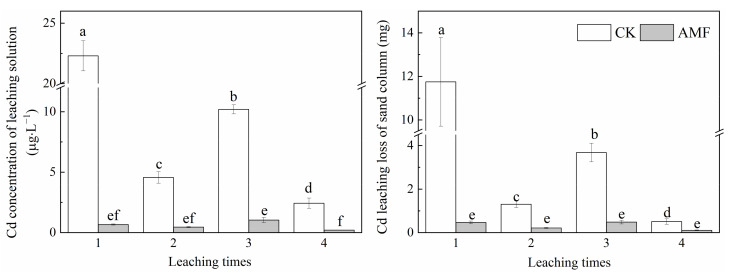
Effect of AMF on the loss of Cd leaching solution from sand column. Error bar indicates standard deviation, n = 4. CK represents the control of non-inoculation, Cd represents cadmium. AMF represents *Funneliformis mosseae* inoculation. Different little letters refer to *p* < 0.05 according to the LSD test.

**Table 1 jof-08-00866-t001:** Effects of AMF on root morphology of maize under 20 mg·L^−1^ Cd concentration stress.

Treatment	Root Length (mm)	Root Surface Area (cm^2^)	Average Root Diameter (mm)	Root Volume (cm^3^)	Root Tip Number (n)	Root Crossings Number (n)
CK	798.8 ± 45.4	38.0 ± 4.7	0.74 ± 0.05	9.4 ± 0.47	51,177 ± 1862	6419 ± 258
AMF	665.3 ± 40.2 **	30.4 ± 2.1 **	0.60 ± 0.10	7.8 ± 0.63 **	45,823 ± 1773 **	4081 ± 366 **

Data in the table are presented as means ± standard deviations (n = 4). Cd represents cadmium, CK represents the control of non-inoculation, AMF represents *Funneliformis mosseae* inoculation. “**” mean *p* < 0.01 according to LSD test.

**Table 2 jof-08-00866-t002:** Effects of AMF on the secretion of low-molecular-weight organic acids in maize under 20 mg·L^−1^ Cd concentration stress.

Treatment	Malonic Acid Content (mg·L^−1^)	Oxalic Acid Content (mg·L^−1^)	Succinic Acid Content (mg·L^−1^)	Malic Acid Content (mg·L^−1^)	Acetic Acid Content (mg·L^−1^)	Citric Acid Content (mg·L^−1^)
CK	4.89 ± 0.22	0.48 ± 0.04	208.2 ± 14.6	19.0 ± 2.7	17.2 ± 2.2	33.9 ± 3.6
AMF	12.43 ± 0.8 **	1.01 ± 0.04 **	320.6 ± 21.7 **	23.4 ± 3.6	0.78 ± 0.11 **	13.6 ± 1.36 **

Data in the table are presented as means ± standard deviations (n = 4). Cd represents cadmium, CK represents the control of non-inoculation, AMF represents *Funneliformis mosseae* inoculation. “**” mean *p* < 0.01 according to the LSD test.

**Table 3 jof-08-00866-t003:** Effect of AMF on Cd distribution in sand column maize system.

Treatment	Cd Adsorption by Sand (mg)	Cd Uptake in Maize (mg)	Cd Leaching Loss (mg)	Cd Adsorption by the Inner Wall (mg)
CK	36.9 ± 0.5 (92.3%)	2.35 ± 0.55 (5.89%)	0.71 ± 0.09 (1.78%)	0.026 ± 0.001 (0.066%)
AMF	39.2 ± 0.6 ** (98.0%)	0.49 ± 0.58 (1.22%)	0.32 ± 0.01 ** (0.80%)	0.004 ± 0.000 ** (0.010%)

Data in the table are presented as means ± standard deviations (n = 4). Cd represents cadmium, CK represents the control of non-inoculation, AMF represents *Funneliformis mosseae* inoculation. “**” mean *p* < 0.01 according to the LSD test. Figures in parentheses are the proportion of Cd in the sand column maize system.

**Table 4 jof-08-00866-t004:** Correlation between root morphology and GRSP, maize Cd accumulation and Cd leaching loss.

Index	Total Root Length	Root Surface Area	Average Root Diameter	Root Volume	Root Tip Number	Root Crossing Number
EE-GRSP	−0.811 **	−0.583	0.059	−0.626	−0.738 *	−0.820 **
T-GRSP	−0.900 **	−0.722 *	0.291	−0.749 *	−0.820 **	−0.863 **
Cd content of shoot	0.914 **	0.754 *	−0.309	0.694 *	0.852 **	0.934 **
Cd content of root	0.921 **	0.791 **	−0.337	0.683 *	0.832 **	0.907 **
Cd uptake by maize shoot	−0.920 **	−0.758 *	0.268	−0.672 *	−0.844 **	−0.860 **
Cd uptake by maize root	−0.848 **	−0.748 *	0.167	−0.702 *	−0.884 **	−0.857 **
Cd leaching loss	0.907 **	0.749 *	−0.322	0.732 *	0.866 **	0.946 **

EE-GRSP represents easily extractable glomalin-related soil protein, T-GRSP represents total glomalin-related soil protein; Cd represents cadmium. “*” and “**” means *p* < 0.05 and *p* < 0.01 according to correlation analysis, respectively. n = 8.

**Table 5 jof-08-00866-t005:** Correlation between root secretion of low-molecular-weight organic acids and GRSP and quartz sand Cd adsorption.

Index	Oxalic Acid	Succinic Acid	Malic Acid	Malonate	Acetic Acid	Citric Acid
EE-GRSP	0.924 **	0.916 **	0.586	0.923 **	−0.892 **	−0.893 **
T-GRSP	0.916 **	0.904 **	0.432	0.912 **	−0.902 **	−0.914 **
Cd adsorption by sand	0.781 **	0.573	0.243	0.735 *	−0.753 *	−0.745 *

EE-GRSP represents easily extractable glomalin-related soil protein, T-GRSP represents total glomalin-related soil protein; Cd represents cadmium. “*” and “**” means *p* < 0.05 and *p* < 0.01 according to correlation analysis, respectively. n = 8.

## Data Availability

Not applicable.

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
