# Peer review of "Arbuscular Mycorrhizal Fungi Reduce Cadmium Leaching from Sand Columns by Reducing Availability and Enhancing Uptake by Maize Roots"

_jof, 2022, doi:10.3390/jof8080866_

Round 1

Reviewer 1 Report

The manuscript "Arbuscular mycorrhizal fungi enhance cadmium adsorption by sand and uptake by maize roots and reduce cadmium leaching loss in a sand column" by Yu and colleagues reports the increased Cd adsorption in of maize plants inoculated by AMF in a sand column system. This is an interesting finding with potential applications in phytoremediation. I have minor comments outlined below: 

Line 43: Please re-word the phrase “Arbuscular mycorrhizal fungi (AMF) is a kind of soil fungi”. AMF are a group of Glomeromycotina fungi that encompass very diverse types of fungi, so the word “kind” makes it seem like there’s only one type which is not correct

Line 272: “…indicating that AMF has a significant effect on Cd loss” should rather say “indicating that AMF has a significant effect on Cd adsorption”

The authors should refrain from using the phrase “extremely significant” (multiple occurrences throughout the manuscript). It is preferable to just state that a difference is statistically significant followed by the p-value.  

The authors show that AM colonization results in decreased plant growth. However, AM colonization also results in increased plant Cd uptake -> could this result in the growth defects? This should be discussed.

Author Response

Point 1: I Line 43: Please re-word the phrase “Arbuscular mycorrhizal fungi (AMF) is a kind of soil fungi”. AMF are a group of Glomeromycotina fungi that encompass very diverse types of fungi, so the word “kind” makes it seem like there’s only one type which is not correct

Response 1: Done. “Arbuscular mycorrhizal fungi (AMF) is a kind of soil fungi" to "Arbuscular mycorrhizal fungi (AMF) are widely distributed soil fungi" in the manuscript

Point 2: Line 272: “…indicating that AMF has a significant effect on Cd loss” should rather say “indicating that AMF has a significant effect on Cd adsorption”

Response 2: Done. We have replaced " indicating that AMF has a significant effect on Cd loss " in line 272 with " indicating that AMF has a significant effect on Cd adsorption".

Point 3: The authors should refrain from using the phrase “extremely significant” (multiple occurrences throughout the manuscript). It is preferable to just state that a difference is statistically significant followed by the p-value.

Response 3: Done. Except for correlations, we express all "extremely significant" as p-values.

Point 4: The authors show that AM colonization results in decreased plant growth. However, AM colonization also results in increased plant Cd uptake -> could this result in the growth defects? This should be discussed

Response 4: In this experiment, although the inoculation of AMF reduced the biomass and increased the absorption of Cd by plants, it also inhibited the transfer of Cd to the shoot and reduced the damage of Cd to the shoot. According to the opinions of reviewers, we discussed it in section 4.2.

Reviewer 2 Report

I have carefully read the manuscript submitted by Zihao Yu et al. I believe the authors must improve the formality to present the document for peer review. The research topic is interesting, but the manuscript in some sentences is hard to read. I have listed several issues that the authors must solve to improve the quality of the manuscript.

The authors must rewrite the abstract section. Essential information regarding the importance of mycorrhizal fungi's influence on the sand's adsorption and limitation of absorption by maize is missing. The authors can shorten the results if necessary.

Introduction: I feel the introduction lacks essential information, such as understanding the main effects of cadmium that can induce plant toxicity and some paragraphs justifying the use of maize instead of other plants.

Lines 15 to 18: the authors must add quantitative data

Line 25: The authors must avoid speculations and focus on the study results.

Line 26: the authors must specify the term mycorrhizal secretion

Line 25:The authors must specify where the Cd is immobilized and where the cadmium is lost.

Line 35: Delete the segment “enhanced at the soil surface, where it is”

Line 43: Replace “is a kind of soil fungi widely distributed in nature” with “are widely distributed soil fungi”

Line 46: Replace “a dense network of hyphae” with “dense hyphal networks”

Line 47. Please rewrite this sentence “contains binding sites that can bind to and fix Cd”. I think fungal hyphae can absorb metals and fix inside and/or outside fungal structures. In addition to GRSP, organic acids can be secreted to form precipitated compounds such as oxalate, malate, and citrate, among others. Such mechanisms must be better explained here in the introduction and can also be discussed if they are related to the study's main findings.

Line 52: this sentence is confusing. Please rewrite: “Meanwhile, AMF significantly affected the Cd distribution in plants, fixed Cd in roots, inhibited the Cd transport to plant stems and leaves, and reduced the Cd uptake in stems and leaves”

Line 60: I feel this segment is confusing. Please rewrite: “Due to the large Cd pool in soil, whether the minimal changes of soil Cd adsorption induced AMF caused an obvious impact on environmental Cd migration, such as plant Cd uptake and Cd leaching loss, is still unclear.

Line 63: duplicate section, please delete: “In view of the fact that most of the previous studies used polluted soil as the matrix, the soil composition is complex, which makes the quantitative analysis of Cd migration in the soil-plant system difficult to control

Line 69: Delete this sentence as it looks like materials and methods: “Therefore, quartz sand was used as the culture medium, Cd solution was injected exogenously, and sand culture experiments were carried out to study the influence of AMF inoculation on maize development, Cd concentration, and Cd migration in the sand column-maize system.

Materials and methods.

Line 84: this paragraph needs to be rewritten. The authors must consider the use of “.”

Sentence in line 94: the sentence must be written in the past form.

Section 2.2: Please, rewrite the sentences with verbs in the past form. Check this issue in all the materials and methods. 

Paragraph at line 105: some concerns: why do the authors use bacteria? At which intervals do the authors apply the nutrient solution? Was it a new solution in each application? Which was the n? Only one plant per treatment? At which time the fungal inoculum was applied? Please clearly specify these issues.

Section 2.3: At which time the root samples were taken? The same concern is about soil. The procedure to obtain GRSP must be clearly described. 

Section 2.5: description of the columns, specific equipment, and main procedure to quantify root exudates must be provided. the authors must also provide the time for sampling

Section 2.6: the “pharmaceutical reagents” must be clearly described and clarified.

Section 2.7.: I did not understand the meaning of the first paragraph in this section

Results section: avoid subjective asseveration such as “dramatically”. Something that “clear” for you probably cannot be evident for me.

In the materials and methods section, the authors state four treatments, but the figures and tables had only two treatments

In section 3.7 the authors indicated that some growth parameters negatively correlate with GRSP and positively with Cd. So, Cd is good for the growth of maize?, and the presence of glomalin in the soil solution (which relates to the presence of AMF in the soil) is negative to improving the growth of maize. This is confusing.  

The discussion section head is missing in the manuscript.

Line 410: this looks like an introduction, not a discussion of your results. The discussion section must focus on your main findings and not on reviewing other studies. For this reason, in some sections, your manuscript looks like a litaterature review. Discussion of your result does not mean repeating the results in the discussion section.

Conclusion: Focus your conclusion on the main findings (not result description) and propose further study areas.

Author Response

Point 1: The authors must rewrite the abstract section. Essential information regarding the importance of mycorrhizal fungi's influence on the sand's adsorption and limitation of absorption by maize is missing. The authors can shorten the results if necessary.

Response 1:Done. We rewrote the abstract, supplemented the relevant quantitative data, revised the effects of AMF on Cd adsorption by sand,Cd uptake by maize, and Cd leaching loss, and further condensed the summary.

Point 2: Introduction: I feel the introduction lacks essential information, such as understanding the main effects of cadmium that can induce plant toxicity and some paragraphs justifying the use of maize instead of other plants.

Response 2: We supplemented the harm of Cd and the mechanism of inducing phytotoxicity, and explained the reasons for using maize in terms of maize biomass, root morphology, Cd tolerance in the manuscript, and supplemented relevant references.

Point 3: Lines 15 to 18: the authors must add quantitative data.

Response 3: Done. We have supplemented quantitative data in Lines 15 to 18, and also in the corresponding discussion section.

Point 4: Line 25: The authors must avoid speculations and focus on the study results.

Response 4: Done. We further summarize the summary part of the abstract, and draw conclusions from the research results.

Point 5: Line 26: the authors must specify the term mycorrhizal secretion.

Response 5: Done. We specified the specific secretion in line 26, instead of replacing it with "mycorrhizal secretion".

Point 6: Line 25:The authors must specify where the Cd is immobilized and where the cadmium is lost.

Response 6: Done. We supplement the fixed position of Cd and the lost position of cadmium in the abstract.

Point 7: Line 35: Delete the segment “enhanced at the soil surface, where it is”

Response 7: Done. We deleted the line 35 "enhanced at the soil surface, where it is"

Point 8: Line 43: Replace “is a kind of soil fungi widely distributed in nature” with “are widely distributed soil fungi”.

Response 8: Done. We have replaced " is a kind of soil fungi widely distributed in nature " in line 43 with " are widely distributed soil fungi ".

Point 9: Line 46: Replace “a dense network of hyphae” with “dense hyphal networks”.

Response 9: We have replaced "a dense network of hyphae" in line 46 with "dense hyphal networks".

Point 10: Line 47. Please rewrite this sentence “contains binding sites that can bind to and fix Cd”. I think fungal hyphae can absorb metals and fix inside and/or outside fungal structures. In addition to GRSP, organic acids can be secreted to form precipitated compounds such as oxalate, malate, and citrate, among others. Such mechanisms must be better explained here in the introduction and can also be discussed if they are related to the study's main findings.

Response 10: Done. We changed “contains binding sites that can bind to and fix Cd” in line 47 to “dense hyphal networks that wraps around soil particles and contains binding sites that can bind to and fix Cd,”, and supplemented the mechanism of the effect of root secreted organic acids on Cd leaching loss below.

Point 11: Line 52: this sentence is confusing. Please rewrite: “Meanwhile, AMF significantly affected the Cd distribution in plants, fixed Cd in roots, inhibited the Cd transport to plant stems and leaves, and reduced the Cd uptake in stems and leaves”

Response 11: Done. We have made changes in accordance with the comments of the reviewers, and the results of the changes are as follows “AMF significantly affected the Cd distribution in plants: Cd was mainly fixed in AMF-inoculated roots, Cd transport to shoot was reduced, resulting in less Cd uptake in shoot.”.

Point 12: Line 60: I feel this segment is confusing. Please rewrite: “Due to the large Cd pool in soil, whether the minimal changes of soil Cd adsorption induced AMF caused an obvious impact on environmental Cd migration, such as plant Cd uptake and Cd leaching loss, is still unclear.

Response 12: Done. We have rewritten this sentence to read as follows: The influence of the minor changes of Cd adsorption in soil induced by AMF on environmental Cd migration, such as plant Cd uptake and Cd leaching loss, is still unclear. 

Point 13: Line 63: duplicate section, please delete: “In view of the fact that most of the previous studies used polluted soil as the matrix, the soil composition is complex, which makes the quantitative analysis of Cd migration in the soil-plant system difficult to control.

Response 13: Done. Based on the comments of the reviewers we have removed the sentence "In view of the fact that most of the previous studies used polluted soil as the matrix, the soil composition is complex, which makes the quantitative analysis of Cd migration in the soil-plant system difficult to control" in the manuscript.

Point 14: Line 69: Delete this sentence as it looks like materials and methods: “Therefore, quartz sand was used as the culture medium, Cd solution was injected exogenously, and sand culture experiments were carried out to study the influence of AMF inoculation on maize development, Cd concentration, and Cd migration in the sand column-maize system.

Response 14: Done. Based on the reviewer's comments, we have removed the sentence "Therefore, quartz sand was used as the culture medium, Cd solution was injected exogenously, and sand culture experiments were carried out to study the influence of AMF inoculation on maize development, Cd concentration, and Cd migration in the sand column-maize system.".

Point 15: Line 84: this paragraph needs to be rewritten. The authors must consider the use of “.”

Response 15: Done. We rewrote Section 2.1 and asked an English language editing company, American Journal Experts, with technical and academic writing to help us modify the language of the paper.

Point 16: Sentence in line 94: the sentence must be written in the past form.

Response 16: Done. We have rewritten this sentence using the past tense as follows: 2.04 g of CdCl2·2.5H2O was dissolved in 10 L of Hoagland nutrient solution prepared a Cd-containing Hoagland nutrient solution with a Cd concentration of 20 mg·L-1.

Point 17: Section 2.2: Please, rewrite the sentences with verbs in the past form. Check this issue in all the materials and methods.

Point 17: Done. We use the past form of verbs to rewrite sentences in the materials and methods section.

Point 18: Paragraph at line 105: some concerns: why do the authors use bacteria? At which intervals do the authors apply the nutrient solution? Was it a new solution in each application? Which was the n? Only one plant per treatment? At which time the fungal inoculum was applied? Please clearly specify these issues.

Point 18: Done. Our previous description was wrong, it should have been "strains" instead of "bacteria", which has been corrected in the manuscript. Since the content of Cd added inside and outside the leaching column is a fixed value, the change of the liquid volume will directly affect the change of the Cd concentration. After the maize emerges, we supplement the nutrient solution in the leaching column every other day to maintain the constant volume in the leaching column, which can achieve the effect of constant Cd concentration and make the stress concentration constant from beginning to end. The volume of the leaching column is limited. If two or more corns are planted in one leaching column, the roots will be entangled and difficult to separate, which will affect the test results. We planted only one corn per leaching column, but to eliminate experimental error, each treatment was replicated four times, with four leaching columns per treatment. We applied the inoculum when filling the sand column with quartz sand, which has been added in the manuscript.

Point 19: Section 2.3: At which time the root samples were taken? The same concern is about soil. The procedure to obtain GRSP must be clearly described. 

Response19: Done. We collected root samples the day after the leaching test. According to the opinions of reviewers, we supplemented the collection time of root samples in the manuscript, the supplementary results are as follows: "After the leaching experiment (maize was grown to 72 days), maize root samples and rhizosphere quartz sand were collected to determine the infection characteristics of AMF.". And we further describe the acquisition process of GRSP in detail.

Point 20: Section 2.5: description of the columns, specific equipment, and main procedure to quantify root exudates must be provided. the authors must also provide the time for sampling.

Response 20: Done.Following the reviewer's suggestion, we have supplemented the description of sampling times, chromatographic columns for quantifying root exudates, specific equipment, and main procedures in the Test Methods for Measuring Root Exudates.

Point 21: Section 2.6: the “pharmaceutical reagents” must be clearly described and clarified.

第2.6节:必须清楚地描述和澄清“药物试剂”。

Response 21: Done.We supplementally describe the specific pharmaceutical reagents used in Section 2.6.

Point 22: Section 2.7.: I did not understand the meaning of the first paragraph in this section

Response 22: Done. We want to explain how the cumulative proportion of Cd in each part of the system is calculated, and this sentence has been rewritten.

Point 23: Results section: avoid subjective asseveration such as “dramatically”. Something that “clear” for you probably cannot be evident for me.

Response 23: Done. We have corrected "dramatically" to "significantly" in the manuscript to avoid ambiguity.

Point 24: In the materials and methods section, the authors state four treatments, but the figures and tables had only two treatments

Response 24: Done. In this experiment, there are two treatments, the control group (CK) and AMF inoculation group (AMF). We modified the materials and methods to meet the consistency.

Point 25: In section 3.7 the authors indicated that some growth parameters negatively correlate with GRSP and positively with Cd. So, Cd is good for the growth of maize?, and the presence of glomalin in the soil solution (which relates to the presence of AMF in the soil) is negative to improving the growth of maize. This is confusing.

Response 24: In the early stage of maize growth, mycelium does not need to absorb fixed nutrients to promote plant growth, while fungi consume most of photosynthetic products (such as carbon sources) to maintain their own growth and development, resulting in insufficient plant demand for carbon sources, thereby inhibiting plant growth . In addition, some studies have also found that if there is sufficient fertility in the growth medium to meet the needs of the root system, it will be difficult for AMF mycelia to obtain nutrients to promote plant growth, thereby inhibiting the growth of host plants. Therefore, AMF will inhibit crop growth under certain conditions, and this experiment also found that inoculation with AMF reduced root morphology and biomass. In addition, AMF inoculation led to an increase in Cd content in maize roots, and the accumulation of toxic Cd ions may mask the positive effects of mycorrhizae and change maize root characteristics, thereby inhibiting maize growth. We have supplemented the discussion section for reasons why AMF limits crop growth.

Point 26: The discussion section head is missing in the manuscript.

Response 26: Done.We have supplemented the title of the discussion section in the manuscript.

Point 27: Line 410: this looks like an introduction, not a discussion of your results. The discussion section must focus on your main findings and not on reviewing other studies. For this reason, in some sections, your manuscript looks like a litaterature review. Discussion of your result does not mean repeating the results in the discussion section.

Response 27: Done. We reworked this sentence. Related discussions are also drawn from our research in the discussion.

Point 28: Conclusion: Focus your conclusion on the main findings (not result description) and propose further study areas.

Response 28: Done. We have changed the "Conclusions" to no longer merely describe the results, in addition, we have suggested further areas of research in the discussion section.

Reviewer 3 Report

This publication presents interesting results of the influence of AMF inoculation on maize development, Cd concentration, and Cd migration in the sand column-maize system. The adopted approach in this study was very interesting since AMF inoculation showed a promising asset to alleviate the deleterious effects of Cd on the environment.

The manuscript was well introduced, and the authors adopted very convincing methods with a consistent discussion of the different obtained results. However, the manuscript needs substantial revisions and an English review to be suitable for publication in Journal of Fungi.

General comments

- Comment 1: The English of this manuscript needs substantial improvements

- Comment 2: The choice of AMF strain, maize, and Cd concentration should be justified.

- Comment 3: Some results are missing.

Other comments

Abstract

1. L27: please delete the numbers next to the keywords.

Introduction

2. L43: please change “is” to “are”.

3. L57-59: this sentence is the same as in lines 63-65, please keep one of the two.

4. L60-63: please rewrite the sentence.

5. L69-74; please rewrite, there is a redundancy.

M&M

6. L78: please write materials without “s”.

7. L83: please add a space between the value and the unit. Please check throughout the manuscript.

8. L91: Funneliformis mosseae, is it a Cd toxicity tolerant strain?

9. L94: please rewrite in passive voice all sentences with imperative tense. The same for L108, 135, 142, 143, 159, 160, 161, ….. please correct this throughout the manuscript.

10. L100: please change “Figure 1 A” to “Figure 1 a”.

11. L108-109: you stated, “Lay 200 g of bacteria on the surface of the AMF treated sand column”. I do not understand, do you mean AMF instead of bacteria? Is it 100 g or 200 g?

12. L110: please change “Figure 1 B” to “Figure 1 b”.

13. L117: specify in the title what is a and b.

14. L114: why did you use 20 mg L-1 as the concentration of Cd solution?

Results

15. L206-207: the results of this part are missing.

16. L262-265: please delete.

17. Table 3: the description of “Cd adsorption by the inner wall” is missing.

18. L279-287: please rewrite by grouping the negative correlations on the one hand and the positive ones on the other hand.

Discussion

19. L309: please add 4. Discussion.

20. L315 and L382: please correct the citation form.

21. L325: You stated “studies” at the beginning of the sentence, and you put one reference. The same in L348; Please check throughout the whole manuscript.

22. L351-357: It is not well discussed, please discuss with more detailed mechanisms. 

Author Response

Point 1: The English of this manuscript needs substantial improvements

Response 1: Done. Thank the reviewers for their review. We asked an English language editing company, American Journal Experts, with technical and academic writing to help us modify the language of the paper.

Point 2: The choice of AMF strain, maize, and Cd concentration should be justified.

Response 2: Done. AMF strains was resistant to Cd stress. Huidan No. 4 maize variety was a low-accumulation variety of Cd, and it is often popularized in heavy metal-contaminated farmland. Cd concentration of 20 mg·L-1 is the concentration screened in the previous pre-experiment, which can produce corresponding stress without killing maize.

Point 3: Some results are missing.

Response 3: Done. We supplement relevant quantitative data in the abstract and results sections.

Point 4: L27: please delete the numbers next to the keywords.

Response 4: Done. We have deleted the numbers next to the keywords according to the comments of reviewers.

Point 5: L43: please change “is” to “are”.

Response 5: Done. We have corrected "is" to "are" in line 43.

Point 6: L57-59: this sentence is the same as in lines 63-65, please keep one of the two.

Response 6: Done. We deleted the sentences of lines 63-65 and kept the sentences of lines 57-59.

Point 7: L60-63: please rewrite the sentence.

Response 7: Done.We have rewritten the sentences on lines 60-63 as follows: The influence of the minor changes of Cd adsorption in soil induced by AMF on environmental Cd migration, such as plant Cd uptake and Cd leaching loss, is still unclear.

Point 8: L69-74; please rewrite, there is a redundancy.

Response 8: Done. We deleted this sentence.

Point 9: L78: please write materials without “s”.

Response 9: Done. We corrected "materials" for L78 to "material".

Point 10: L83: please add a space between the value and the unit. Please check throughout the manuscript.

Response 10: Done. We checked the full manuscript and added a space between the value and the unit.

Point 11: L91:Funneliformis mosseae,is it a Cd toxicity tolerant strain?

Response 11: Done. Funneliformis mosseae have strong tolerance to Cd and has a certain application value in the remediation of Cd-polluted soil. We have supplemented the manuscript with the reasons for the use of this strains and the relevant references.

Point 12: L94: please rewrite in passive voice all sentences with imperative tense. The same for L108, 135, 142, 143, 159, 160, 161, ….. please correct this throughout the manuscript.

Response 12: Done. We rewrite all sentences with imperative tense in passive voice, and asked an English language editing company with technical and academic writing to help us modify the language of the paper.

Point 13: L100: please change “Figure 1 A” to “Figure 1 a”.

Response 13: Done. We have changed “Figure 1 A” to “Figure 1 a” in the manuscript.

Point 14 L108-109: you stated, “Lay 200 g of bacteria on the surface of the AMF treated sand column”. I do not understand, do you mean AMF instead of bacteria? Is it 100 g or 200 g?

Response 14: Our previous description was wrong, it should have been "strains" instead of "bacteria", which has been corrected in the manuscript.

Point 15 L110: please change “Figure 1 B” to “Figure 1 b”.

Response 15: Done. We have changed “Figure 1 B” to “Figure 1 b” in the manuscript.

Point 16 L117: specify in the title what is a and b.

Response 16: Done. We specify what a and b are in the title, and add the following results: a: Schematic of the leaching device; b: Schematic diagram of leaching column.

Point 17 L114: why did you use 20 mg L-1 as the concentration of Cd solution?

Response 17: Cd concentration of 20 mg·L-1 is the concentration screened in the previous pre-experiment, which can produce corresponding stress without killing maize.

Point 18 L206-207: the results of this part are missing.

Response 18: This part is about the mycorrhizal infection characteristics of maize under the stress of 20 mg·L-1 Cd. There are only three data: AMF infection rate, spore number and hypha density, which are only used to illustrate the symbiotic relationship between maize and AMF. It is of little significance to make a chart, so we directly explained it in the manuscript.

Point 19 L262-265: please delete.

Response 19: This part is the effect of AMF on Cd adsorption by sand, which is one of the focuses of this paper. We think section 3.5 should be kept。

Point 20 Table 3: the description of “Cd adsorption by the inner wall” is missing.

Response 20: We introduced "Cd adsorption by the inner wall" in 2.6 yuan. Because the highlight of this experiment is that AMF is mediated by mycorrhizal secretion, increasing Cd adsorption by sand and Cd uptake in plants can help to reduce Cd leaching loss, which further indicates that AMF has potential important ecological functions in passivating soil Cd pollution and reducing Cd loss. This process has little relation with Cd adsorption on the inner wall, so it is not further described.

Point 21 L279-287: please rewrite by grouping the negative correlations on the one hand and the positive ones on the other hand.

Response 21: Done. We have changed the order of correlation analysis as required.

Point 22 L309: please add 4. Discussion.

Response 22: Done. We have supplemented the title of the discussion section in the manuscript.

Point 23 L315 and L382: please correct the citation form.

Response 23: Done. We have reworked the citation format for L315 and L382 according to the journal format.

Point 24 L325: You stated “studies” at the beginning of the sentence, and you put one reference. The same in L348; Please check throughout the whole manuscript.

Response 24: Done. We modified "studies" for L325 and L348 to "study" and detected similar problems in the whole manuscript.

Point 25 L351-357: It is not well discussed, please discuss with more detailed mechanisms.

Response 25: Done. We have rewritten the discussion part of L351-357, and further discussed the effects of nutrient absorption and Cd stress on organic matter.

Round 2

Reviewer 2 Report

Dear Editor

The authors strongly enhanced the manuscript presentation. However, I feel that further work is necessary for the discussion.

The title must be modified. I suggest something like:

Arbuscular mycorrhizal fungi reduce cadmium leaching from sand columns by reducing availability and enhancing uptake by maize roots. 

This title is reasonable if, in your analisis, you tested this enhancing cadmium uptake by maize roots.

The first 16 lines are theoretical aspects (good for introduction, not discussion). Some short contextualization is required in the paragraphs but not a literature review. Still, I feel that the first paragraph (and some segments) are not linked to the study results, thereby being completely unnecessary in this section.  Some wording such as extremely significantly negatively” must be simplified; negatively is enough. I suggest avoiding the wording  “may be”. When you use this term looks like speculations. In line 425 you did not test heavy metals, only cadmium. Finally, I dislike the conclusion as you only listed the main results. I prefer the summary paragraph of the discussion with some modifications:

 “This study showed that AMF inoculation plays a significant role in reducing the leaching of Cd from contaminated soil (CONFIRM IF YOU USED SOIL OR SAND). The subtle changes (SPECIFY WHICH CHANGES) in the soil (OR SAND) Cd were closely related to environmental Cd migration (I DO NOT KNOW IF YOU EVALUATED ENVIRONMENTAL CADMIUM MIGRATION IN YOUR EXPERIMENTS, YOU NEED TO BE SPECIFIC TO THE PERFORMED TESTS) (Cd uptake by plants and solution Cd loss). However, further studies are necessary to determine the influence of AMF on the leaching of cadmium from contaminated soil under different environmental conditions.

I hope the authors accept this suggestion necessary to improve the manuscript quality. Best regards.

Author Response

Point 1: The title must be modified. I suggest something like: Arbuscular mycorrhizal fungi reduce cadmium leaching from sand columns by reducing availability and enhancing uptake by maize roots. This title is reasonable if, in your analisis, you tested this enhancing cadmium uptake by maize roots.

Response 1: Done. Based on the suggestions of reviewers, we have revised the title and replaced "Arbuscular mycorrhizal fungi enhance cadmium adsorption by sand and uptake by maize roots and reduce cadmium leaching loss in a sand column" with " Arbuscular mycorrhizal fungi reduce cadmium leaching from sand columns by reducing availability and enhancing uptake by maize roots".

Point 2: The first 16 lines are theoretical aspects (good for introduction, not discussion). Some short contextualization is required in the paragraphs but not a literature review. Still, I feel that the first paragraph (and some segments) are not linked to the study results, thereby being completely unnecessary in this section. Some wording such as extremely significantly negatively” must be simplified; negatively is enough. I suggest avoiding the wording “may be”. When you use this term looks like speculations.

Response 2: Done. We simplified the first 16 lines of the discussion and changed "extremely significantly negatively" to "significantly negatively". In addition, we corrected "may be" to "is" to enforce certainty.

Point 3: In line 425 you did not test heavy metals, only cadmium.

Response 3: Done. We corrected "heavy metals" here to "Cd", after which we checked the full text and corrected all similar errors.

Point 4: Finally, I dislike the conclusion as you only listed the main results. I prefer the summary paragraph of the discussion with some modifications: This study showed that AMF inoculation plays a significant role in reducing the leaching of Cd from contaminated soil (CONFIRM IF YOU USED SOIL OR SAND). The subtle changes (SPECIFY WHICH CHANGES) in the soil (OR SAND) Cd were closely related to environmental Cd migration (I DO NOT KNOW IF YOU EVALUATED ENVIRONMENTAL CADMIUM MIGRATION IN YOUR EXPERIMENTS, YOU NEED TO BE SPECIFIC TO THE PERFORMED TESTS) (Cd uptake by plants and solution Cd loss). However, further studies are necessary to determine the influence of AMF on the leaching of cadmium from contaminated soil under different environmental conditions.

Response 4: Done. We revised the conclusion section as requested by the reviewers, deleted the repeated results , changed "soil" to "sand", and additionally mention that the subtle change of Cd in soil was because of the subtle changes of Cd adsorption by sand. However, it should be noted that in the sand—maize column system there are only two fates of Cd: uptake by maize and leaching lost. Therefore, we think that the environmental Cd migration in this experiment was due to Cd uptake by plants and solution Cd loss.

Reviewer 3 Report

The authors satisfied all the raised comments. I endorse the publication of the article.

Author Response

Thank you for your review and approval.